# Infrared Clinical Enamel Crack Detector Based on Silicon CCD and Its Application: A High-Quality and Low-Cost Option

**DOI:** 10.3390/jimaging7120259

**Published:** 2021-12-02

**Authors:** Yuchen Zheng, Min-Hee Oh, Woo-Sub Song, Ki-Hyun Kim, In-Hee Shin, Min-Seok Kim, Jin-Hyoung Cho

**Affiliations:** 1Department of Orthodontics, School of Dentistry, Chonnam National University, Gwangju 61186, Korea; zycscola@gmail.com; 2Department of Orthodontics, School of Dentistry, Dental 4D Research Institute, Dental Science Research Institute, Chonnam National University, Gwangju 61186, Korea; dentoh0423@hanmail.net; 3Medical Photonics Research Center, Korea Photonics Technology Institute, Gwangju 61005, Korea; wsong@kopti.re.kr (W.-S.S.); kimkh@kopti.re.kr (K.-H.K.); ihshin@kopti.re.kr (I.-H.S.); 4Department of Oral Anatomy, School of Dentistry, Dental Science Research Institute, Chonnam National University, Gwangju 61186, Korea; greatone@chonnam.ac.kr

**Keywords:** cracked-tooth syndrome, enamel crack, dentistry imaging, near-infrared ray, transmission, aging

## Abstract

Enamel cracks generated in the anterior teeth not only affect the function but also the aesthetics of the teeth. Chair-side tooth enamel crack detection is essential for clinicians to formulate treatment plans and prevent related dental disease. This study aimed to develop a dental imaging system using a near-IR light source to detect enamel cracks and to investigate the relationship between anterior enamel cracks and age in vivo. A total of 68 subjects were divided into three groups according to their age: young, middle, and elderly. Near-infrared radiation of 850 nm was used to identify enamel cracks in anterior teeth. The results of the quantitative examination showed that the number of enamel cracks on the teeth increased considerably with age. For the qualitative examination, the results indicated that there was no significant relationship between the severity of the enamel cracks and age. So, it can be concluded that the prevalence of anterior cracked tooth increased significantly with age in the young and middle age. The length of the anterior enamel cracks tended to increase with age too; however, this result was not significant. The silicon charge-coupled device (CCD) with a wavelength of 850 nm has a good performance in the detection of enamel cracks and has very good clinical practicability.

## 1. Introduction

Tooth cracks have been investigated by many clinicians for a long time. In 1954, Thoma and Goldman defined “fissured fracture” as a crack that existed in the crown of a tooth, which could be found only in enamel or in enamel and dentin [1]. Clinically, even a negligible enamel crack can cause irreversible damage to the dental tissue, such as dental decay, tooth sensitivity, and tooth splitting. Furthermore, enamel cracks on teeth are easily stained, which can affect dental aesthetics.

As a result of difficulty in detection due to invisibility, enamel cracks are not easy to find initially. According to the previous studies, transillumination, quantitative light-induced fluorescence technology with a digital camera (QLF-D), and swept-source optical coherence tomography (SS-OCT) were all declared as effective methods to detect dental cracks [2,3,4]. However, the studies mentioned above were carried out in vitro. In clinical treatment, if the patient’s enamel cracks can be detected in accurately real time and by chair side, this will be helpful for the dentist’s diagnosis and treatment plan.

Since tooth enamel has high transparency at the wavelength of near-infrared rays (Near-IR), near-infrared ray is very suitable for the detection of enamel cracks. In 2014, Fried et al. showed that Near-IR at 1300 nm is valuable for detecting cracks [5]. However, the detection of 1300 nm wavelength requires the use of the expensive InGaAs system, which is not friendly to the popularization of Near-IR tooth crack detection device in dental clinics. Jones et al. studied the transillumination of interproximal caries lesions using a 830 nm light source and showed that sound enamel and the simulated lesion are detected more significantly at 830 nm than in visible light [6]. In this study, a chair-side detection device with a silicon CCD camera at 850 nm wavelength was used to detect the patient’s enamel cracks in vivo in real time.

The etiology of enamel cracks is a hot topic of investigation. However, the etiology is not singular but multifactorial. Sometimes, dental treatment itself can be a risk factor for tooth cracking [7,8,9,10,11,12]. On the other hand, for untreated natural teeth, trauma caused by accident, parafunctional habit, occlusal interference, and excessive bite forces are considered risk factors for enamel cracks [7,9,13]. Some researchers have found a significant relationship between tooth cracking and aging [8,14]. However, only a few studies have focused on enamel cracks in the anterior teeth, which play an important role in both function and aesthetics.

The early detection of enamel cracks is imperative. From the point of view of the treatment plan, for those small, superficial enamel cracks, clinicians remove the compromised portion and then restore with composite, pinned amalgam, or appropriate cast restoration. For those cracks that involve the pulp or even extend to the alveolar bone, orthodontic band and cast metal restorations may be required, and severe cases may even require tooth extraction [10].

In order to explore the effectiveness of 850 nm wavelength imaging in the detection of enamel cracks, the present study compared three groups of patients with different age and tried to find the relationship between enamel cracks and aging.

## 2. Materials and Methods

### 2.1. Subjects

All the subjects in the present study were aged between 18 and 65 years old and recruited by the Chonnam National University Dental Hospital (CNUDH) orthodontic department. For aesthetic reasons, the subjects’ anterior teeth were sampled according to the two inclusion criteria: (1) subjects who had no history of orthodontic treatment; and (2) subjects whose anterior teeth had no more than four teeth undergoing endodontics, prosthodontics, or restoration treatment. The subjects with more than four abfraction, missing, severely rotated, and severely defected teeth in the anterior teeth were excluded.

All the subject data were collected and recorded anonymously with full patient consent and approval from the Institutional Review Board of CNUDH (IRB code: CNUDH-2019-013). A power analysis using G Power software (version 3.1.3; Franz Faul University, Kiel, Germany) determined that a sample size of 21 subjects per group would provide power of 80% to detect significant differences with an 0.8 effect size and alpha value of 0.05. The effect size was determined based on a previous study, which evaluated the number of cracks in various age groups [15]. Initially, 81 subjects were recruited in the present study. After further screening, eight subjects were excluded from the study because of their orthodontic treatment history. Finally, another five subjects were excluded because of their restorations or crowns on more than four anterior teeth. As a result, a total of 68 subjects were divided into three groups according to their age: “young” (18 ≤ age ≤ 34, N = 22), “middle” (35 ≤ age ≤ 50, N = 23), and “elderly” (age ≥ 51, N = 23). The average age of each group was 26, 43, and 57, respectively. The number of male and female was 35 and 33, respectively.

Before the examination, the subjects were asked questions by the examiner to confirm that they met the inclusion criteria. Once verified, the subjects were asked to gargle, wear a mouth retractor, and adjust the sitting position to obtain clear and uncovered images. All the detections were conducted by one author (Y.Z.).

### 2.2. Detection Method

In the present study, all the tooth images were obtained intraorally by a homemade dental imaging system with Near-IR light source (Figure 1a). A mini-LED that emits Near-IR light in the 850 nm band was attached on a flexible printed circuit board (PCB), and silicone was injected into the PCB to fabricate a light source in the form of a mouthpiece. As shown in Figure 1b, the light source module consists of a total of 32 mini-LEDs, and the maximum intensity is 300 mW. In addition, the output of the light source can be adjusted in 5 steps through a controller. The dental imaging unit consists of a camera (Guppy Pro IEEE1394, Allied Vision, Stadtroda, Germany), a lens (67714 VIS-NIR, Edmund optics, Barrington, IL, USA), a touch panel, and a mini-PC (UP Squared Pentium Quad Core, eMMC, Santa Clara, CA, USA). It can also be rotated to take images of various tooth parts.

Since the direction of Near-IR transillumination is from the lingual surface to the buccal surface, only the enamel cracks on the buccal or lingual surface of the anterior teeth were included for analysis. Figure 1c shows the positional relationship between the subject and the CCD camera. At least three images of the maxillary anterior teeth were captured for each subject from different directions: frontal, left, and right images. Similar images were obtained for the mandibular teeth. To obtain a clear image of the crack, the examiner sometimes adjusted the brightness of the LED lights on the tray to get different pictures in the same direction for comparison. Several intraoral photos were taken by another examiner after crack detection, which were used for confirming the enamel crack. The intraoral photo was taken using a Nikon D7200 SLR camera (Nikon Corporation, Tokyo, Japan), which was utilized as a reference to identify the enamel cracks.

#### 2.2.1. Quantitative Evaluation

Of the included teeth, 52 teeth were excluded because of previous prosthodontics treatment, restorations, crown breakage, or severe rotation. Finally, a total of 764 teeth from 68 subjects were included in the present study, of which 247, 267, and 250 teeth belonged to the young, middle, and elderly age groups, respectively. An average of 11.24 anterior teeth for each subject were selected for evaluating. In order to clearly determine the relationship between the number of enamel cracks in teeth and age, all teeth with enamel cracks were divided into four categories according to the following classification: Type 0, no crack on the tooth, *n* = 0; Type I, only one crack on the cracked tooth, *n* = 1; Type II, two cracks on the cracked tooth, *n* = 2; Type III, more than two cracks on the cracked tooth, *n* ≥ 3 (Figure 2, *n* indicates the number of cracks). When identifying the cracks, external photos were taken for reference. The software Photoshop (Version 20.0.0 ×64, Adobe Photoshop CC 2019, Adobe Inc., San Jose, CA, USA) was utilized to adjust the brightness, contrast, and definition of the images to identify the crack more reliably.

#### 2.2.2. Qualitative Evaluation

To intuitively reflect the difference in the severity of enamel cracks, all cracks were classified according to the crack length as follows (Figure 3). In Class 1, the length of the crack is no more than 1/2 of the crown length along the crack direction. In Class 2, the length of the crack is more than 1/2 of the crown length along the crack direction but not more than 3/4. In Class 3, the length of the crack is more than 3/4 of crown length along the crack direction. Two evaluators (Y.Z., M.O.) performed both the quantitative and qualitative classification.

### 2.3. Statistical Analysis

All the sample data in the present study conformed to the normal distribution. The data were analyzed using the IBM SPSS statistical software (version 22.0 SPSS Inc., Chicago, IL, USA). An intraclass correlation coefficient (ICC) and independent sample test were used to test the inter-examiner reliability. Multivariate logistic regression was used to determine the existence of differences among the three age groups according to the tooth type. With regard to the qualitative aspect, the multivariate logistic regression was also used to determine the existence of differences in the crack severity among the three age groups. A significance level of 0.05 was considered statistically significant. The confidence level of 95% was used for statistical analysis.

## 3. Results

The intraclass correlation coefficients for the quantitative and qualitative evaluation were 0.804 and 0.924, which showed excellent reliability. The results of the independent sample test for the quantitative and qualitative evaluation were 0.136 and 1, respectively, which suggested that there were no significant differences between the two examiners.

After all the tooth evaluations, a total of 291 enamel cracks were detected in the 764 teeth. The largest number of cracks was detected in the upper central incisor of the elderly group (22.3%), which was followed by the lower central incisor of the elderly (14%) and middle group (11.3%). The number of cracks detected in the upper lateral incisor of the young group and lower canine of the elderly group was the least, with only one crack detected in both (Table 1).

As mentioned above, all teeth were classified into four types based on the existing number of cracks, indicating the frequency of the cracks. Typical dental imaging for each age group is shown in Figure 2. In all age groups, enamel cracks are clearly visible in the imaging at the 850 nm wavelength. When contrasted with the naked eye under visible light, enamel cracks under Near-IR were more likely to be found.

The number of Type 0 teeth (healthy teeth) decreased with age. In the other three categories, the number of cracked teeth increased with age. The multivariate logistic regression analysis revealed that the prevalence of Type I (*p* = 0.028) and Type II (*p* = 0.006) cracked teeth in the young group was significantly lower than that in the middle group. The prevalence of Type III cracked teeth in the young group was not significantly different compared to that in the middle group (*p* = 0.197).

The prevalence of Type I to III cracked teeth in the young group and the middle group was significantly lower than that in the elderly group (Figure 4a, Table 2).

According to the odds ratio analysis, the prevalence of Type III cracked teeth in the elderly group was 32.01 times higher than that in the young group, and this was the largest difference in cracked tooth prevalence between the young and the other two age groups. The smallest difference in prevalence was for Type I cracked teeth between the young and the middle group, the prevalence being 1.85 times higher in the middle group than that in the young group. Even though the odds ratio analysis showed that the prevalence of Type III tooth in the middle group was 4.25 times higher than that of the young group, the difference was not significant. The odds ratio for the comparison between the middle and elderly group showed that the prevalence of Type III teeth in the elderly group was 7.53 times higher than that in the middle group, while for the other two types, the prevalence was only approximately two times higher (Table 1).

About the classification of enamel cracks, the Class 1 crack in the elderly group was the most common crack found in the study, which was followed by the Class 2 crack in the elderly group and the Class 1 crack in the middle group. The multivariate logistic regression results showed no significant differences between the young and middle group and between the young and elderly group with respect to all three classes.

Odds ratio analysis indicated that compared with the young group, the risks of Class 2 and 3 cracks in the middle group were 1.37 and 1.41 times higher, respectively. However, the risk of a Class 1 crack in the middle group was 0.94 times higher than that in the young group. The risks of suffering from all the three kinds of cracks in the elderly group were higher than those in the young group but not significantly (Figure 4b, Table 3).

## 4. Discussion

In several previous studies that also used Near-IR to detect enamel cracks or defects, the Near-IR at 1300 nm or 1310 nm was selected; the enamel was almost transparent at this wavelength, which was more conducive to the detection of tooth defects [16,17,18,19]. In the present study, we found that 850 nm was also good at displaying cracks with great clarity and accuracy. Studies have shown that although the imaging contrast at the 830 nm wavelength is not as high as that can be achieved at 1310 nm, its performance is acceptable in a clinical environment in view of the low-cost characteristics of silicon CCDs [6]. The Near-IR detection device used in this study has a wavelength of 850 nm and can perform real-time detection in vivo. The present study showed that this device has an acceptable performance and quite good clinical applicability compared with the expensive InGaAs system.

This study is the first study to classify the severity of enamel crack based on the axial length of the crack, which was divided into three categories according to its proportion to the length of the clinical crown. In this study, Near-IR was used to detect enamel cracks in vivo, and only the cracks on buccal or lingual surfaces of the anterior teeth were detected. In related studies, some authors have also tried to classify the cracks on the tooth. Jun et al. used QLF-D to classify the cracks into four categories, according to their depth: sound, out-half, and inner-half cracks of enamel, and cracks reaching the dentin–enamel junction [3]. They detected cracks not only on the buccal and lingual surfaces but also on the occlusal surface of the teeth. In this study, we analyzed the length of the enamel crack as we believe that the length of the enamel crack is essential for assessing the severity of cracks in anterior teeth.

Several studies that also examined the relationship between tooth cracks and age grouped patients into age groups [8,20]. Hiatt analyzed the fractured teeth of 64 patients, and the average age of the subjects was 44 years old, which was similar to the average age of the 68 patients in this study (42 years old) [21]. The results showed that most cracks were found in the elderly group, followed by the middle and young groups. Compared to other age groups, teeth in the elderly age group have been used for longer, suggesting that their risk of cracking due to abrasion and accidental trauma greatly increased. Although we tried to exclude traumatic teeth by observing and questioning patients when selecting samples, some injuries that were not obvious and not easily felt by patients, which might lead to tooth cracks, were observed.

From the quantitative perspective, the statistical results showed that there was a significant increase in all types of teeth (except for Type III in the young group) in both the middle group and the elderly group compared to the young group. Additionally, significant differences in all types of teeth were also found between the middle and elderly groups. These results indicate that the number of cracked teeth increased significantly with aging. In comparison, the number of cracked teeth between the middle and elderly group tended to increase more rapidly than that between the young and middle group. This result indicates that the cracked teeth increased extremely significantly after the age of 50. The reason that Type III teeth did not differ significantly between the young and middle group may be because of the small number of Type III teeth detected: only one in the young group and four in the middle group. Previous studies have yielded similar results. Cameron examined 50 incomplete fractured teeth, and 72% of them were originally from patients over the age 50 years old [22]. Twelve years later, his follow-up study confirmed the results again. He detected 102 incomplete fractured teeth, 58% of which were over 50 years old [23]. However, it is essential to note that both studies only analyze the prevalence of posterior teeth and do not mention anterior teeth. Cameron believed that the reason for this result was that teeth became more brittle and fragile as they aged [22].

Some studies did not match the current findings. Hiatt examined 100 incomplete fractured teeth from 64 patients and found that the incidence of fractured teeth declined rapidly after the age of 50 years old [21]. However, the study included both restored and untreated teeth, so it was likely that early restorations protected the teeth from cracking.

In terms of qualitative analysis, the severity of enamel cracks increased with age, but this trend was not significant. For Class 1 cracks, the middle group even showed a decreasing trend compared to the young group. In this study, the crack length was taken as the standard to measure the severity of the enamel crack, and the results showed that the crack length did not increase significantly with age. Several previous studies have used crack depth as a measure of severity. Turssi et al. studied 355 premolars and classified the severity of the tooth crack into four categories according to their crack depth [24]. The results showed that the age of teeth was moderately correlated with the severity of cracks. This suggests that the tooth cracks may become significantly deeper rather than longer with age.

Previous studies have shown that parafunctional habits may be an influential factor of enamel crack such as bruxism [7,22]. However, because most subjects of this study did not know whether they had bruxism or not, in order not to affect the accuracy of the results, bruxism was not analyzed and discussed as a variable in this study. Another limitation of this study is that the device used in this study cannot detect the depth of the crack. As mentioned earlier, depth is also an indicator of enamel crack severity and an important factor in the treatment plan. In the follow-up studies, it is necessary to consider bruxism as one of the influencing factors for enamel cracks. According to the exclusion criteria, some teeth were excluded before evaluation, which resulted in there not being equivalent numbers of tooth types included in the evaluation. Due to the detection direction, this device can only effectively detect enamel cracks in the anterior teeth. Future research should consider how to detect and analyze cracks in the posterior teeth in vivo.

Enamel cracks are the initial cause of many oral diseases and the beginning of structural damage to teeth. In this study, a high-quality and inexpensive silicon CCD detector was used to detect enamel cracks at a wavelength of 850 nm and obtained clear images. However, for deeper tooth cracks, InGaAs cameras with good performance around 1300 nm can obtain better images. In the future design of this device, how to detect the cracks of the posterior teeth more easily and how to detect the deeper tooth cracks are issues that need to be considered.

## 5. Conclusions

In this study, Near-IR was used to detect the enamel cracks in the anterior teeth of Korean people in vivo, and the relationship between age and enamel cracks of the subjects was analyzed quantitatively and qualitatively. It could be concluded that the prevalence of anterior cracked tooth increased significantly with age and increased more rapidly after 50 years old. The length of anterior enamel cracks tended to increase with age, but this result was not statistically significant. In addition, the silicon CCD with a wavelength of 850 nm showed a good performance in the detection of enamel cracks and has good clinical practicability.

## Figures and Tables

**Figure 1 jimaging-07-00259-f001:**
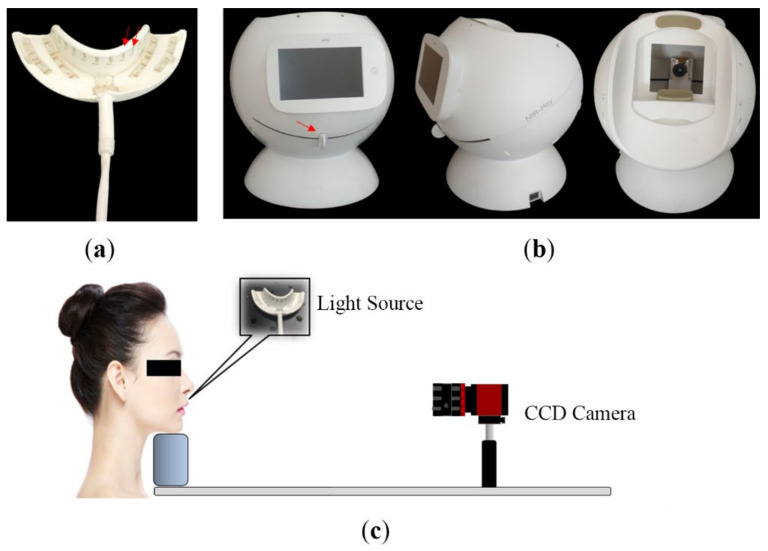
Device and method of detection. (**a**) Rubber tray, which could emit 850 nm Near-IR in the mouth to highlight the cracks. Red arrows indicate the LED illuminant, whose brightness can be adjusted from level 1 to 5, with 5 being the brightest. (**b**) Near-infrared dental and periodontal imaging device with a screen on the examiner’s side for image capturing and a CCD camera inside for image acquiring. Red arrow indicates a control lever, which is connected to the camera to adjust its direction during the detection process. (**c**) Schematic diagram of the patient position during detection.

**Figure 2 jimaging-07-00259-f002:**
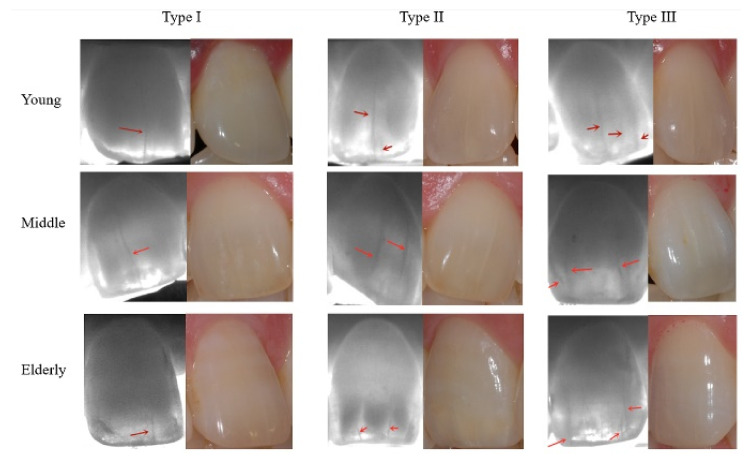
The three most representative types of teeth in the three age groups. The red arrow points out the enamel cracks. The high contrast of the enamel cracks in the near-infrared image can be clearly seen.

**Figure 3 jimaging-07-00259-f003:**
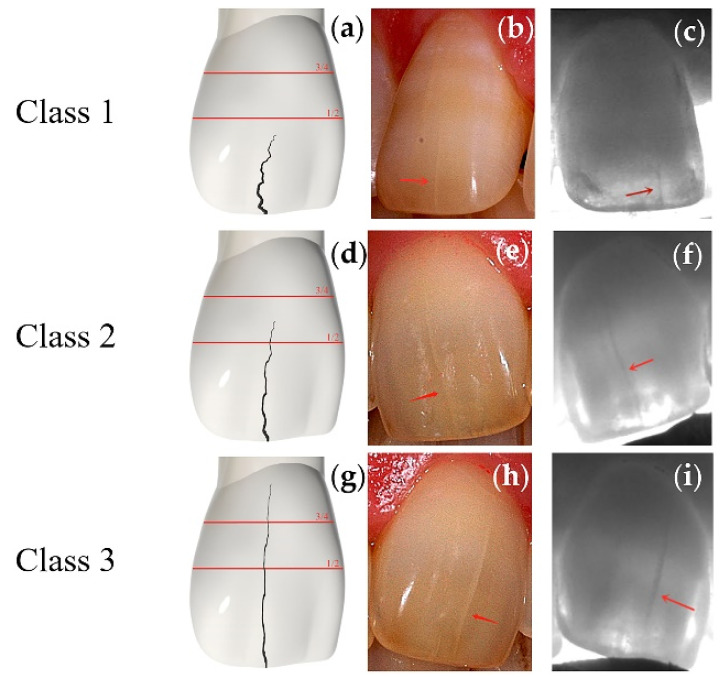
Simulated images, intraoral photos, and 850 nm Near-IR images for indicating the differences among the classifications. Red arrows indicate the cracks. (**a**) A simulated Class 1 central incisor crack. (**b**) An intraoral photo of a Class 1 crack on the maxillary central incisor under natural light. (**c**) The image of the tooth in Figure 3b under Near-IR. (**d**) A simulated Class 2 central incisor crack. (**e**) The intraoral photo of a Class 2 crack on the maxillary central incisor under natural light. (**f**) The image of the tooth in Figure 3e under Near-IR. (**g**) A simulated Class 3 central incisor crack. (**h**) An intraoral photo of a Class 3 crack on the maxillary central incisor under natural light. (**i**) The image of the tooth in Figure 3h under Near-IR.

**Figure 4 jimaging-07-00259-f004:**
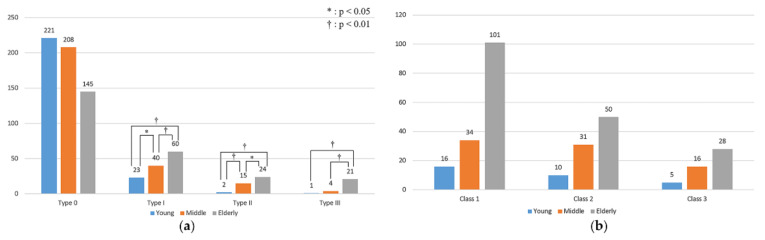
The distribution of different tooth types and tooth classifications among the three groups. (**a**) Tooth type distribution. The value of the ordinate indicates the number of cracked teeth. (**b**) Enamel crack classification distribution. The value of the ordinate indicates the number of different classes of cracks.

**Table 1 jimaging-07-00259-t001:** Distribution of the enamel cracks in different types of teeth among different age groups.

		Young	Middle	Elderly	Total
Maxilla	Central incisor	14 (4.8%)	29 (10.0%)	65 (22.3%)	108
	Lateral incisor	1 (0.3%)	8 (2.7%)	11 (3.8%)	20
	Canine	4 (1.4%)	6 (2.0%)	21 (7.2%)	31
Mandible	Central incisor	7 (2.4%)	33 (11.3%)	41 (14%)	81
	Lateral incisor	3 (1.0%)	4 (1.4%)	28 (9.6%)	35
	Canine	2 (0.7%)	1 (0.3%)	13 (4.5%)	16
Total		31	81	179	291

**Table 2 jimaging-07-00259-t002:** Multivariate logistic regression result of qualitative analysis among three age groups.

Group		Middle			Elderly		
		Sig ^1^	OR ^2^	CI ^3^	Sig	OR	CI
Young	Type I	0.028	1.848	1.07 to 3.19	0	3.976	2.35 to 6.72
	Type II	0.006	7.969	1.8 to 35.27	0	18.29	4.26 to 78.57
	Type III	0.197	4.25	0.47 to 38.34	0.001	32.007	4.26 to 240.55
	Type I	-	-	-	0.001	2.152	1.37 to 3.38
Middle	Type II	-	-	-	0.016	2.295	1.16 to 4.53
	Type III	-	-	-	0.000	7.531	2.53 to 22.40

^1^ Sig, significance. Significance analysis from multivariate logistic regression, *p* < 0.05. ^2^ OR, odds ratio. ^3^ CI, confidence interval, 95%.

**Table 3 jimaging-07-00259-t003:** Multivariate logistic regression results of qualitative analysis of the crack class.

Group		Sig ^1^	OR ^2^	CI ^3^
Middle	Class 1	0.864	0.936	0.44 to 1.99
	Class 2	0.471	1.366	0.58 to 3.19
	Class 3	0.542	1.41	0.47 to 4.26
Elderly	Class 1	0.198	1.563	0.97 to 3.09
	Class 2	0.602	1.238	0.55 to 2.77
	Class 3	0.539	1.387	0.49 to 3.94

^1^ Sig, significance. Significance analysis from multivariate logistic regression, *p* < 0.05. ^2^ OR, odds ratio. ^3^ CI, confidence interval, 95%.

## Data Availability

The data presented in this study are not publicly available due to restrictions in the research ethics.

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
