# Peer review of "Infrared Clinical Enamel Crack Detector Based on Silicon CCD and Its Application: A High-Quality and Low-Cost Option"

_2313-433X, 2021, doi:10.3390/jimaging7120259_

Round 1

Reviewer 1 Report

I would like to thank the Authors for the submission of this interesting research, that provide clear alternative to the detection of enamel cracks. The research has been well conducted and the manuscript is clearly written. However some minor revisions are required.

Kindly collocate the references numeration before punctuation, as written in the guidelines and not in the middle of sentences.

Plagiarism has been checked and a very low level of similarity was detected (2%).

INTRODUCTION

  • Lines 45-46: Kindly add the abbreviation of near-infrared ray here, since it is the first time you mentioned it.
  • In order to complete the background of your research, kindly add a short paragraph on treatment choice for enamel cracks.

MATERIAL AND METHODS

  • Lines 127-130: kindly add the description images (fig 3) of type 0, 1, 2 and 3, as done for the qualitative analysis.

DISCUSSION

  • Kindly add a comment on the limitations and drawback of the research, for example the inability of the device to detect the depth of the enamel crack, that could be an important factor in the treatment plan.

Reviewer 2 Report

Please refer the attachment.

Author Response

Following your suggestion, our manuscript has been revised by an English native-speaking colleague.

Reviewer 3 Report

Thank you for the opportunity to review an article on an interesting and important topic. 
Introduction:
The introduction sufficiently thoroughly explains the theoretical basis of the issue. In addition, the last paragraph states the purpose of the study.

Materials and methods
The authors provided inclusion criteria. Were there any exclusion criteria as well?
The method of the study was adequately described by the authors, with the use of figures. 
Results
Figure 4 is numbered 1. Thus there are two figures 1. Also, due to its small size, it is difficult to read. 
The discussion and conclusions describe the topic well, and the conclusions correspond to the stated objectives. 
In the reviewer's opinion, the paper in its current form can be published in MDPI JIMAGING. 
